# Hydrochemical Evolution of Groundwater in a Typical Semi-Arid Groundwater Storage Basin Using a Zoning Model

**Mingqian Li** [1,2] , **Xiujuan Liang** [1,2,*], **Changlai Xiao** [1,2], **Yuqing Cao** [1,2] **and Shuya Hu** [1,2]

1   Key Laboratory of Groundwater Resources and Environment, Ministry of Education, Jilin University, Changchun 130026, China
2   College of New Energy and Environment, Jilin University, Changchun 130021, China
*   Correspondence: xjliang@jlu.edu.cn

**Abstract:** Groundwater guarantees water resources and ecological environment security in semi-arid areas. Studying the chemical evolution of groundwater in semi-arid areas is of great significance to regional ecological environment protection and water resources management. The water storage basin is not only a space for groundwater storage and movement but also a place for water–rock–gas interaction and elemental migration, dispersion, and enrichment. Due to its unique climate and geological environment, the semi-arid water storage basins have different hydrochemical environments, forming a zonal hydrogeochemical character. In this study, a typical semi-arid water storage basin (west of Jilin Province) is taken as an example, through the cross section of the recharge–runoff–excretion zone. A three-level hydrogeochemical zoning model is constructed to reveal the hydrogeochemical evolution of the area. The model is divided into three layers from bottom to top. The first layer shows the geological and hydrogeological conditions, including the topography, lithology, geological time, and hydrodynamic characteristics of the study area. The second layer represents the hydrogeochemical processes, divided into the recharge zone, runoff zone, and discharge zone in the horizontal direction according to the hydrodynamic environment and hydrochemistry type. The hydrogeochemical action gradually changes from lixiviation to cation exchange, evaporation, and concentration, as the landform plays a key role in hydrochemistry formation in the discharge area. The third layer gives the characteristics of the groundwater chemical components, including chemistry type, total dissolved solids, main anion and cation, and characteristic element F. Qualitative and quantitative characterizations of hydrochemistry evolution by reverse simulation and hydrodynamic, hydrochemical and thermodynamic indicators are given.

**Keywords:** hydrogeochemical evolution; zoning model; semi-arid area; groundwater storage basin

## 1. Introduction

Groundwater is an essential part of the global water cycle, serving as the link between the lithosphere, the biosphere, and the atmosphere. The groundwater circulation involves the movement of solvents (water molecules) and solutes [1]. The formation of groundwater chemical characteristics is the result of the long-term interaction between groundwater and the surrounding environment, with a complex exchange of substances and energy, and the continuous migration, dispersion, and enrichment of elements. At present, the research on hydrogeochemical evolution mainly focuses on genesis and source analysis, the migration and enrichment of the feature elements (e.g., Fe, F, As), quantitative chemical simulation, groundwater pollution remediation, and the intrinsic relationship with groundwater circulation [2–6]. The study of regional hydrogeochemical evolution can help us

reshape the regional hydrogeological history, clarify the origin and formation of groundwater, reveal the interaction mechanism between groundwater and the environment, and provide a scientific basis for environmental protection and water resource management [7,8].

It is well known that changes in and the distribution of climate, vegetation, and animals on the earth exhibit zonation to a certain extent, and the chemical composition of groundwater is no exception. The essence of hydrochemistry zoning is that the differentiation of the hydrogeochemical environment and the zoned scale is determined by the scale of the chemical environment. The scale can be a large continental structural scale. For example, in China, from southeast to northwest, groundwater of bicarbonate-type water, low in total dissolved solids (TDS), gradually turns into a complex $SO_4^{2-}$ type water, finally to become high-TDS $Cl^-$ type water [9]. The scale can also be micro-geomorphic, such as groundwater in hills with high altitude and deep groundwater buried depth, which is generally $HCO_3$-type water with low TDS, while groundwater in a low-lying fulje with a shallow groundwater buried depth is typically $HCO_3 \cdot Cl$ or Cl-type water with high TDS [10]. Usually, the large regional zoning depends on the climatic conditions, while the small-scale zoning depends on the topographical geological conditions.

The diversity of research subjects and zoning criteria determines the diversity of the zoning model. The authors of [11] divided the groundwater in Datong Basin into five hydrochemical zones: leaching zone, converging zone, enriching zone, oxidizing zone, and reducing zone according to the hydrogeochemical environment. The authors of [12] divided the piedmont plain of North China into the piedmont alluvial area, central alluvial-lake plain, and coastal plain, based on the topography. The authors of [13] divided the coastal area of Turkey into a freshwater holding area and a seawater intrusion area, according to the distribution of groundwater salinity. The authors of [4] divided the groundwater genesis of the Yinchuan Plain into a leaching zone, evaporation-concentration zone, and an evaporation-mixing zone, according to the degree of influence of natural and human irrigation activity on groundwater chemistry. The authors of [14] divided the Polish glacial-buried valley aquifer into a precipitation recharge area, lateral runoff recharge area, and leakage recharge area, based on different groundwater sources. However, hydrochemistry zonation can be inconspicuous because of local hydrogeochemical environmental anomalies [15,16].

These studies indicate that the chemical composition and formation of groundwater cannot be analyzed in isolation from the perspective of just chemistry. The evolutionary law must be revealed from the perspective of the interaction between water and the environment. However, most studies have only qualitatively analyzed groundwater chemical zoning. The authors of [1], taking the hydrogeological entity "hydrological geological storage unit" as the research object, used the combination of chemical thermodynamics and chemical kinetics to illustrate new hydrogeochemical differentiation theory in the water–gas–rock equilibrium system, and a new expression of Darcy's law established by hydrochemistry indicators and a new method for calculating hydrogeological parameters and groundwater age based on hydrochemistry data was proposed. According to the theory, for typical hydrogeological water storage structural units, the difference in the scale, direction, and intensity of tectonic movement determines the difference between topography and stratum lithology, causing the groundwater dynamical characteristics, circulation conditions, and intensity of the water–rock–gas interaction to change from the edge to the center of the structural unit, which will affect the direction and progress of the chemical evolution of groundwater [1]. As a result, the chemical composition of the groundwater is spatially differentiated according to a certain law and exhibits horizontal zonation [1,4].

The western part of Jilin is a typical semi-arid water storage basin with less precipitation, strong evaporation, fragile ecological environment, and human life, depending heavily on groundwater [17]. Therefore, this study considers the western part of Jilin as the study area and establishes a zoning model of groundwater chemical evolution under the guidance of hydrogeochemical differentiation theory. The formation of major ions, and the migration and accumulation of trace elements were simulated by PHREEQC v3.4.0 (United States Geological Survey), combined with the interpretation of various parameters (e.g., hydrodynamic parameters, ion ratio coefficient, and thermodynamic index). The hydrogeochemical evolution of typical semi-arid water storage basins is further qualitatively

and quantitatively described. For other semi-arid water storage basins, this study of groundwater chemistry evolution has a guiding significance.

## 2. Materials and Methods

### 2.1. Study Area

The study area is located in the southwestern part of the Songnen Plain in northeastern China, including Baicheng City, and western part of Songyuan City (Figure 1), with a total area of 41,140 km$^2$ and a population of about 4,406,300. The study area is in the transition zone between the humid monsoon area and the arid inland (Figure 1a) and is a typical semi-arid inland basin. The annual average temperature is 3–6 °C, with average annual precipitation of 400–500 mm, mainly concentrated from June to September, with an average annual evaporation of 1500–2000 mm (Figure 2).

The surface water system in the basin is not well developed with only the Taoer River flowing through it, and the northern part of the basin is bordered by the Nen River and Songhua River. As shown in Figure 1b, the terrain is high in the east, south, and west, and low in the middle and north. The northwest is the hilly area of the Mt. Daxing'an fold belt, the southwest is the Songliao watershed, the central part is a broad alluvial and lacustrine plain, and the east is the tableland. The Quaternary strata cover the entire area, but the thickness varies, generally from 40 to 80 m [10]. The main landform types are the western Mountainous area, Taoer alluvial fan, Songnen low plain, tableland and valley plain (Figure 1c).

The study area is a typical semi-closed water storage basin structure, and the hydrogeological conditions are mainly controlled by the geomorphology and climate. The phreatic water and pore-fissure confined water occur in the Quaternary strata and the former is the object in this study. In the Taoer alluvial fan, the phreatic aquifer is composed of middle and upper Pleistocene gravel, with a thickness of 10–40 m and a groundwater buried depth of 10–20 m, which is recharged by meteoric water infiltration, river leakage, and lateral runoff from the western mountainous area [10]. In the low plain, the upper Pleistocene silty sand is dominant in the phreatic aquifer, followed by loess loam, with a thickness of 3–20 m and a groundwater buried depth less than 5 m [10,18]. The low and flat terrain of the low plain makes it conducive for the phreatic aquifer to receive rainfall infiltration recharge. Furthermore, the lateral runoff from the alluvial fan and tableland is also a source of recharge. Evaporation and artificial mining are the main methods of discharge. Controlled by topography, phreatic water flows from the east, south, and west to the middle and north, and flows into the Nen River and Songhua River at the northeastern boundary (Figure 3).

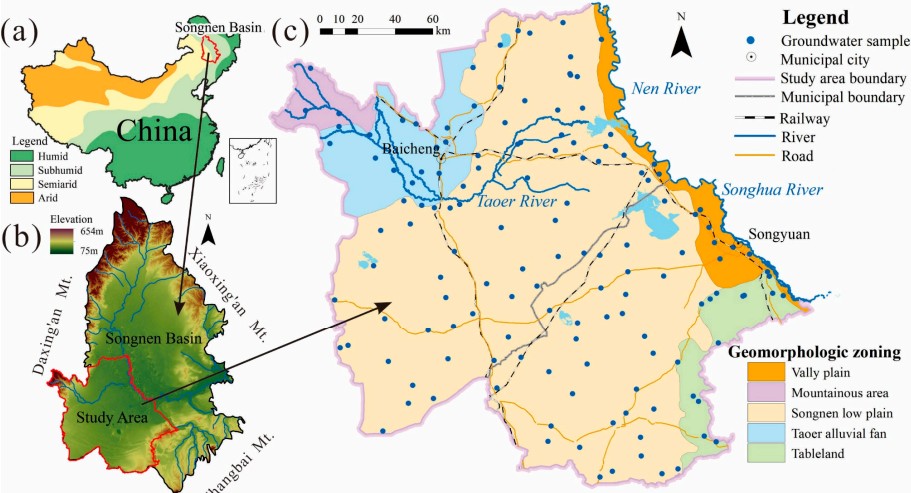

**Figure 1.** Location of the study area. (**a**) Songnen Plain in China and China climate zonation, (**b**) the terrain of the Songnen Plain and the study area in the Songnen Plain, (**c**) landform division and sampling point distribution map of the study area.

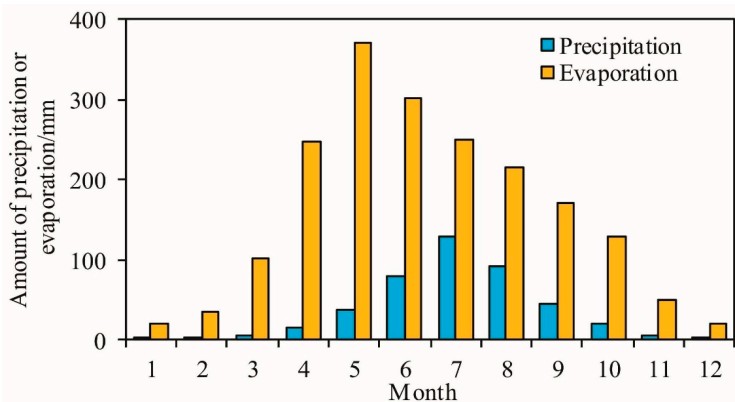

**Figure 2.** Monthly mean value of precipitation and evaporation in the study area from 1951 to 2017.

*2.2. Sample Collection and Analysis*

Groundwater sampling and water level survey were conducted from 3 to 24 November 2017. A total of 142 phreatic water samples (Figure 1c) were collected regularly from pumping wells with depths of 10–30 m (all water samples were from the Quaternary phreatic aquifer). The sampling was carried out 5–10 min after the start of the pumping, to drain the stagnant water in the suction pipe. The water samples were filtered through a 0.45 μm membrane and added to a 350 mL polyethylene plastic bottle, which was rinsed three times with deionized water. Three bottles of water were taken from the same well, and 10% $HNO_3$ was added to one of the bottles to make the water sample pH less than 2 for cation analysis. Water temperature and pH were tested on-site using the calibrated HANNA (HI99131) portable pH/temperature analyzer, and the alkalinity was determined on-site by the Gran titration.

The water samples were tested by Pony Testing International Group in Changchun within one week. The water quality testing method is based on the Standard Examination Methods for Drinking Water (GB5750-2006). The main cations ($K^+$, $Na^+$, $Ca^{2+}$, $Mg^{2+}$, $Fe^{3+}$) were tested by inductively coupled plasma atomic emission spectrometry (ICP-AES). $F^-$, $Cl^-$, $NO_3^-$, and $SO_4^{2-}$ were measured by an ion chromatograph. $NO_2^-$ and $NH_4^+$ were determined by the ultraviolet and visible spectrophotometers, and the TDS was measured by an electric blast drying oven and an electronic analytical balance (vapor-drying method). The reliability of the water sample analysis data was checked by the relative error of the anion and cation milliequivalent, and the error of all water samples was less than 5%. Statistics of groundwater chemical data in each zone are presented in Table 1.

**Table 1.** Statistics of groundwater chemical data in each hydrogeochemical zone (in mg/L, figures in parentheses represent mean values).

| Parameters | Western Recharge Area (n = 19) | Western Runoff Area (n = 46) | Central Discharge Area (n = 43) | Eastern Runoff Area (n = 13) | Eastern Recharge Area (n = 21) |
|---|---|---|---|---|---|
| pH | 7.2–7.8 (7.57) | 7.1–8.2 (7.66) | 7.1–8.3 (7.69) | 7.3–8 (7.68) | 7.5–8 (7.73) |
| TDS | 188–710 (413.29) | 362–1020 (577.81) | 529–3500 (1109.55) | 271–773 (495.42) | 183–527 (355.67) |
| $K^+$ | 0.29–1.94 (0.78) | 0.29–2.32 (1.04) | 0.52–29.3 (2.19) | 0.43–1.99 (0.83) | 0.17–0.89 (0.48) |
| $Na^+$ | 13.1–51.5 (26.65) | 23.2–221 (84.4) | 28.2–710 (199.66) | 13.3–156 (59.39) | 6.07–48.1 (18.18) |
| $Ca^{2+}$ | 31.6–115 (72.02) | 19.9–141 (73.03) | 19.5–341 (101.34) | 44.3–103 (69.08) | 30.6–166 (83.34) |
| $Mg^{2+}$ | 7.69–42.7 (18.07) | 12–79.2 (32.05) | 8.89–182 (57.61) | 13.1–82.3 (32.68) | 4.57–21.3 (12) |
| $Fe^{3+}$ | 0.01–1.39 (0.17) | 0.01–43.2 (3.71) | <0.01–21.5 (1.88) | <0.01–0.65 (0.26) | <0.01–1.35 (0.18) |
| $Cl^-$ | 9–84.9 (25.4) | 2.99–313 (40.09) | 7.98–638 (148.4) | 7.08–121 (40.77) | 3.79–68 (21.28) |
| $SO_4^{2-}$ | 12–56.7 (32.93) | 1.78–178 (41.3) | 0.88–336 (77.85) | 1.73–52.2 (17.67) | 1.17–61.9 (17.78) |
| $HCO_3^-$ | 126–530 (257.35) | 210–1170 (512.19) | 289–1480 (598.07) | 276–848 (457.83) | 146–374 (271.87) |
| $F^-$ | 0.3–0.68(0.58) | 0.53–1.74 (1.15) | 1.03–3.07 (1.62) | 0.6–2.5 (1.29) | 0.21–0.86 (0.63) |
| $NO_3$-N | 0.59–38.9 (15.63) | <0.01–57 (3.03) | <0.01–210 (28.9) | <0.01–5.17 (1.16) | <0.01–30.4 (10.52) |
| $NO_2$-N | <0.001–0.03 (0.01) | <0.001–0.06 (0.01) | <0.001–0.16 (0.02) | <0.001–0.19 (0.03) | <0.001–0.26 (0.03) |
| $NH_4$-N | 0.03–0.34 (0.11) | 0.03–2.95 (0.61) | 0.02–3.38 (0.4) | 0.03–0.54 (0.22) | 0.03–0.37 (0.11) |

### 2.3. Construction of the Hydrogeochemical Zoning Model

Simple and representative models are indispensable in disciplinary research. Therefore, the hydrogeological entity "groundwater storage basin" that exists in reality is selected as the research object [1]. The earth's crust, under the geological action forced by internal and external power, forms folds, faults, and various landforms, which determines the distribution framework of the mountains and plains, and the dynamic environment of the water flow determines the spatial distribution of the particles inside the frame. The loosely deposited substance of different particle sizes is regularly differentiated and deposited, thereby controlling the spatial distribution of the aquifer and aquitard, and the structure of the groundwater system.

In a typical water storage structure, from the recharge area to the runoff area and discharge area, the permeability of the aquifer, the groundwater flow rate, the alternating characteristics of the groundwater circulation, the geological and hydrogeochemical environment, and the water–gas–rock interaction strength exhibit changes on a regular basis, leading to a clear zoning of hydrogeochemical characteristics, especially in semi-arid areas where precipitation and evaporation differ greatly.

Therefore, a hydrogeochemical zoning model of a semi-arid water storage basin based on typical profiles can be constructed (Figure 4). The real groundwater chemical formation process is approached by three levels: geological and geomorphological conditions, hydrochemical processes, and water chemical composition characteristics. AutoCAD2018 is used to describe and build the zoning model.

### 2.4. Hydrogeochemical Simulation

The quantitative analysis of groundwater chemical formation (e.g., rock dissolution and precipitation, evaporation, cation exchange) in the basin was carried out using hydrogeochemical reverse simulation. Considering that the water chemical evolution is mainly reflected in changes in the macroscopic components ($Na/Mg/Ca$ and $HCO_3/Cl/SO_4$) and trace components ($F$), the representative mineral calcite/dolomite/gypsum and halite of the typical water storage structure system were selected as the simulated mineral phase. In addition, the following factors are considered: (1) clay minerals are widely distributed and cation exchange is prone to occur, (2) pore-phreatic aquifer is an open system, as $CO_2$ participates in water–rock interaction, and (3) the strong evaporation in the basin, which can be indicated by the escape of $H_2O$ from the groundwater system. The possible mineral phases also include $CaX_2/NaX$, $CO_2$, and $H_2O$. According to the groundwater flow direction and hydrogeological conditions, two simulation paths of $I_1$–$I_5$ and $II_1$–$II_4$ are selected (Figure 3). Reverse simulation and saturation index calculations were performed using PHREEQC 3.4.0 (United States Geological Survey) [19] developed by the USGS (U.S. Geological Survey).

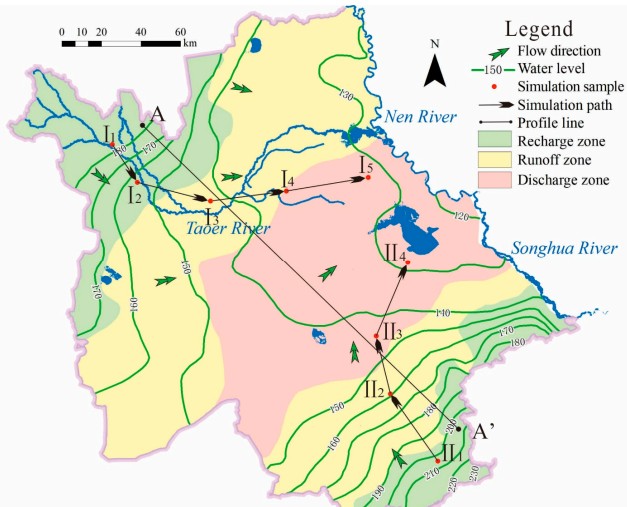

**Figure 3.** Groundwater chemical zonation/flow field and reverse chemical simulation path in the study area.

## 3. Results

### 3.1. Hydrogeochemical Zoning Model

The variation in phreatic water chemical composition is consistent with the changes of the regional hydrogeological conditions. From the piedmont to the center of the basin, as the terrain becomes smooth, the groundwater chemical type gradually transitions from $HCO_3$-Ca(Ca·Mg) to $HCO_3$-Ca·Na(Na·Ca) to $HCO_3$-Na or $HCO_3$·Cl-Na type, showing good "geomorphology-hydrogeochemical process-groundwater chemical composition" zonal character. The range of recharge zones, runoff zones and discharge zones can be divided based on the geological features and the groundwater dynamic environment of the basin, combined with the distribution of groundwater chemistry types (Figure 3).

A hydrogeochemical zoning model (Figure 4) was established by the profile A-A' (Figure 3) passing through two main recharge areas and discharge areas in the basin. The model was divided into three layers from bottom to top. The first layer represented geological and hydrogeological conditions, including the topography, lithology, geological time and hydrodynamic characteristics of the study area. The second layer represented hydrogeochemical processes, which was divided into the recharge zone/runoff zone and the discharge zone in the horizontal direction according to the hydrodynamic environment and hydrochemistry types, and the hydrogeochemical action gradually changes from lixiviation to cation exchange/evaporation and concentration. The third layer represented the characteristics of groundwater chemical components, including chemistry type, TDS, main anion and cation and the characteristic element F. Another analogous zoning model showing hydrogeochemical evolution controlled by micro-topography was built by profile P-P' (Figure 5).

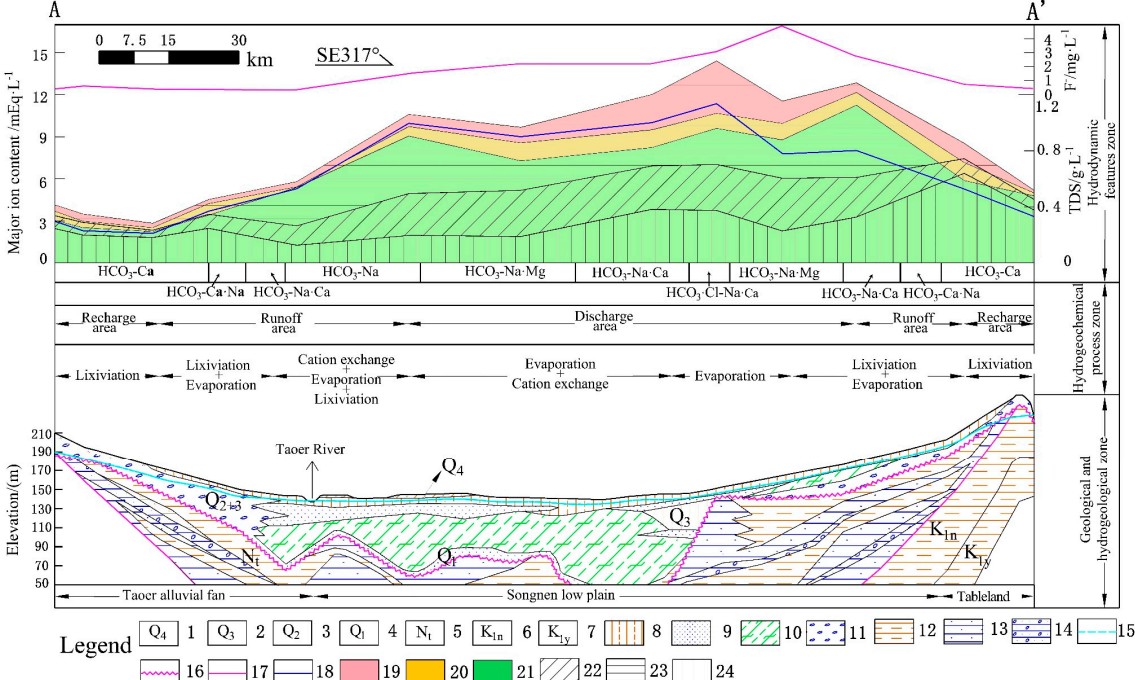

**Figure 4.** Zoning model for the hydrogeochemical evolution of the study area. The color of stratum: Blue for aquifer, green for aquitard and yellow for aquiclude. Legend: (1) Holocene, (2) upper Pleistocene, (3) middle Pleistocene lower Cretaceous Nenjiang Formation, (4) lower Pleistocene, (5) Neogene system Taikang Formation, (6) lower Cretaceous Nenjiang Formation, (7) lower Cretaceous Yaojia Formation, (8) loess loam, (9) fine sand, (10) silty loam, (11) sandy gravel, (12) mudstone, (13) sandstone, (14) glutenite, (15) unconfined water level, (16) angular unconformity interface, (17) $F^-$, (18) TDS, (19) $HCO_3^-$, (20) $SO_4^{2-}$, (21) $Cl^-$, (22) $Mg^{2+}$, (23) $Na^+$, (24) $Ca^{2+}$.

## 3.2. Hydrogeochemical Simulation

The results of the hydrogeochemical simulation are shown in Table 2. In the alluvial fan recharge zone ($I_1$–$I_2$), $CO_2$ enters the groundwater, all minerals are in a dissolved state, and evaporation and cation exchange are relatively weak. When the groundwater flows through the runoff zone ($I_2$–$I_3$), calcite precipitates first (Equation (1)), with a precipitation amount of 0.94 mmol/L, while other minerals are dissolved. Evaporation is enhanced with the evaporation multiple being 1.67. Cation exchange is significant, and $Ca^{2+}$ in groundwater is replaced by the $Na^+$ adsorbed by the rock and soil (Equation (3)); the exchange amount is 1.31 mmol/L (calculated as $Na^+$). When entering the discharge area ($I_3$–$I_4$), dolomite (Equation (2)) and calcite precipitate at the same time and decarbonation occurs; $CO_2$ escapes from the groundwater; the evaporation and cation exchange is remarkable.

In the discharge area, calcite and dolomite precipitate while the fluorite/gypsum and halite are always in a dissolved state. Reverse cation exchange occurs (Equation (4)), and the exchange amount is 1.22 mmol/L (calculated as $Na^+$). The mineral dissolution and precipitation of path II is similar to that of path I. The recharge area is dominated by mineral dissolution; calcite and dolomite precipitate simultaneously at the runoff zone while $CO_2$ escapes from the groundwater. The evaporation is enhanced but the cation exchange is not significant; the reverse cation exchange occurs in the discharge zone.

$$Ca^{2+} + 2HCO_3^- \rightarrow CaCO_3\downarrow + CO_2\uparrow + H_2O \tag{1}$$

$$Ca^{2+} + Mg^{2+} + 4HCO_3^- \rightarrow CaMg(CO_3)_2\downarrow + 2CO_2\uparrow + 2H_2O \tag{2}$$

$$Ca^{2+}(l) + 2NaX(s) \rightarrow 2Na^+(l) + CaX_2(s) \tag{3}$$

$$2Na^+(l) + CaX_2(s) \rightarrow Ca^{2+}(l) + 2NaX(s) \tag{4}$$

**Table 2.** The result of hydrogeochemical simulation.

| Items | | Pathway I (Alluvial Fan → Low Plain) | | | | Pathway II (Tableland → Low Plain) | | |
|---|---|---|---|---|---|---|---|---|
| | | $I_1$–$I_2$ | $I_2$–$I_3$ | $I_3$–$I_4$ | $I_4$–$I_5$ | $II_1$–$II_2$ | $II_2$–$II_3$ | $II_3$–$II_4$ |
| Evaporation Multiple | | 1.00 | 1.67 | 1.61 | 1.20 | 1.03 | 2.14 | 1.17 |
| Mineral dissolution and precipitation [1] | Calcite | 0.64 | −0.94 | −0.61 | −0.67 | 0.49 | −1.06 | −1.96 |
| | Fluorite | 0.01 | 0.001 | 0.01 | 0.01 | 0.03 | 0.002 | 0.003 |
| | Dolomite | 0.33 | 0.41 | −0.74 | −1.09 | 0.22 | −0.46 | −1.51 |
| | Gypsum | 0.13 | 0.02 | 0.31 | 0.21 | 0.06 | 0.51 | 0.01 |
| | Halite | 0.15 | 0.25 | 0.47 | 3.26 | 0.08 | 1.32 | 0.53 |
| | $CO_2(g)$ | 1.01 | 0.85 | −4.05 | −3.57 | 0.65 | −0.14 | −1.03 |
| | $H_2O(g)$ [2] | 0.00 | −37.27 | −33.88 | −11.28 | −1.91 | −63.3 | −9.25 |
| Cation Exchange [3] | $CaX_2$ | −0.23 | −0.65 | −0.91 | 1.22 | −0.25 | −0.14 | 0.43 |
| | NaX | 0.46 | 1.31 | 1.82 | −2.44 | 0.50 | 0.28 | −0.86 |

[1] The positive value in the column of mineral dissolution and precipitation represents dissolution, and the negative value indicates the precipitation; [2] The unit of $H_2O$ is mol/L, and the rest is mmol/L; [3] A positive value of NaX represents the reaction of Equation (3), and a negative value indicates the reaction of Equation (4).

## 3.3. Qualitative/Quantitative Indicator

The qualitative and quantitative characterization of zonation is carried out from two levels as hydrogeological and hydrochemical conditions; the results are shown in Table 3. Hydrogeological conditions determine the water flow environment, which is the key to control the hydrochemical evolution of groundwater. It is qualitatively described by lithology and recharge conditions, and quantitatively analyzed by elevation, groundwater buried depth, and hydrodynamic index (hydraulic gradient, coefficient of permeability, $\gamma Cl^-/\gamma Ca^{2+}$).

The hydrochemical conditions are quantitatively characterized by the coefficients of the ion ratio, the saturation index, and the slope of $\gamma(Na^+ + K^+ - Cl^-)$ and $\gamma(Ca^{2+} + Mg^{2+}) - (SO_4^{2-} + HCO_3^-)$. The coefficients of the ion ratio can be used to confirm the genesis of groundwater or the source of groundwater

chemical composition [6,20]; the mineral saturation index is usually understood as a thermodynamic index describing the ability of groundwater to dissolve a mineral, which can be used to judge the reaction state between water and minerals [1,21]. The relationship of $\gamma(Na^+ + K^+ - Cl^-)$ and $\gamma(Ca^{2+} + Mg^{2+}) - (SO_4^{2-} + HCO_3^-)$ can be used to indicate the extent of cation exchange; if the two are linear and the slope($k_{cation\ exchange}$) is close to $-1$, the cation exchange is significant [6]. The qualitative and quantitative description of each indicator in a different area (Figures 6 and 7, and Table 3) explain the evolution and formation process of groundwater chemistry.

**Table 3.** Qualitative and quantitative indexes of hydrogeochemical zonation.

| Indexes | | Zones | | | | |
|---|---|---|---|---|---|---|
| | | Western Recharge Area | Western Runoff Area | Central Discharge Area | Eastern Runoff Area | Eastern Recharge Area |
| Geomorphology | Landform | Taoer alluvial fan | | Low plain 140–150 (hills) 130–140 (fulje) | | Tableland |
| | Elevation/m | 143–210 | 135–150 | | 155–230 | 173–290 |
| Qualitative description of aquifer | Lithology of aquifer | Gravel/sand gravel, with 0–2 m sandy loam covering on the surface | The upper part is sandy loam and the lower part is fine sand | The upper part is loess loam while lower part is fine sand or sandy loam | The upper part is loess loam or fine sand and the lower part is gravel | The northern part is a gravel and the southern part is Neogene sandstone |
| Quantitative description of aquifer | Hydraulic gradient/$10^{-3}$ | 0.667 | 0.318 | 0.109 | 0.624 | 1.111 |
| | Buried depth/m | 7–15 | 4–12 | <5 | 4–15 | 10–40 |
| | Permeate coefficient/m·day$^{-1}$ | 50–200 | 3–5 | 0.2–0.3 | 3–4 | 15–20 |
| | $\gamma Cl^-/\gamma Ca^{2+}$ | 0.32 | 0.38 | 0.86 | 0.36 | 0.21 |
| Cation exchange | $k_{cation\ exchange}$ | −1.341 ($R^2$ = 0.610) | −0.796 ($R^2$ = 0.845) | −1.048 ($R^2$ = 0.863) | −0.660 ($R^2$ = 0.648) | −1.939 ($R^2$ = 0.738) |
| Ion ratio | $\gamma Ca/\gamma Na$ | 3.273 | 1.285 | 0.931 | 2.143 | 6.089 |
| | $\gamma Mg/\gamma Na$ | 1.406 | 0.915 | 0.774 | 1.271 | 1.442 |
| | $\gamma Ca/\gamma Mg$ | 2.419 | 1.444 | 1.265 | 1.597 | 4.439 |
| | $\gamma Cl/\gamma HOC_3$ | 0.210 | 0.218 | 0.380 | 0.159 | 0.230 |
| | $\gamma SO_4/\gamma HCO_3$ | 0.210 | 0.230 | 0.230 | 0.050 | 0.130 |
| | $\gamma Na/\gamma Cl$ | 2.187 | 4.320 | 2.620 | 2.620 | 2.154 |
| | $\gamma Ca/\gamma SO_4$ | 5.000 | 3.760 | 2.990 | 5.440 | 6.020 |
| | $\gamma Ca/\gamma HCO_3$ | 0.919 | 0.495 | 0.645 | 0.492 | 1.130 |
| Saturation index | $SI_{calcite}$ | 0.120 | 0.539 | 0.631 | 0.393 | 0.425 |
| | $SI_{dolomite}$ | −0.124 | 0.965 | 1.231 | 0.638 | 0.245 |
| | $SI_{gypsum}$ | −1.692 | −1.875 | −1.686 | −2.209 | −1.851 |
| | $SI_{halite}$ | −7.744 | −7.169 | −6.454 | −7.516 | −7.775 |
| | $SI_{fluorite}$ | −1.459 | −1.093 | −0.910 | −0.728 | −1.173 |
| Hydrochemical characteristics | Geochemical reaction | Mainly lixiviation | Mainly lixiviation, cation exchange and evaporation enhanced | Mainly evaporation and cation exchange | Mainly lixiviation and evaporation enhanced | Mainly lixiviation |
| | TDS | 0.2–0.6 | 0.4–1 | 0.5–3.5 | 0.3–0.8 | 0.2–0.5 |
| | Characteristic element: F | 0.3–0.7 | 0.5–2 | 1–3 | 0.6–2.5 | 0.2–0.9 |
| | Hydrochemical type | $HCO_3$-Ca | $HCO_3$-Ca·Na (Na·Ca) | $HCO_3$-Na (Na·Mg)$HCO_3$·Cl-Na | $HCO_3$-Ca·Na (Na·Ca) | $HCO_3$-Ca |

## 4. Discussion

### 4.1. Hydrogeochemical Zoning Model

#### 4.1.1. Geological and Hydrogeological Conditions

The prototype of the Songnen Plain, the Delonggang Penelplane, was formed in the middle of the Middle Pleistocene, and the neotectonic movement formed it into the Songnen Plain known today [10,22]. The study area is located in the southern part of the Songnen Plain as a water-storage tectonic basin that opens to the north (Figure 1b). According to the difference in the form and intensity of the new tectonic movement, the study area is divided into five landforms: western mountainous area, the Taoer alluvial fan, the Songnen low plain, the tableland, and the valley plain (Figure 1c).

The Taoer alluvial fan is the transition zone from the uplift zone of the Mt. Daxing'an fold belt to the fault basin, and the upstream slope is steep (15‰), while the downstream is slow (7‰), with an elevation of 143–210 m. The aquifer consists of the middle Pleistocene alluvial gravel and sand gravel, which is overlying 0–2 m sandy loam, and the aquifer has good permeability, with a permeability

coefficient of 50–200 m/day. The central low plain is a subsidence area, and the ground undulates in the form of microwaves, with an elevation of 130–150 m. It has long been accepted through sedimentary and accumulational evidence that the aeolian landform developed well. The upper part of the aquifer is loess loam while the lower part is fine sand or sandy loam, and the thickness is 5–25 m with low permeability (the permeability coefficient is only 0.2–0.3 m/day), and the underlying layer is thick silty clay. Due to the ground uplift, the eastern tableland has formed a ridge-like watershed with steep slopes and strong ground cutting, with an elevation of 173–290 m. The aquifer is mainly gravel in the north and Neogene sandstone in the south.

The buried depth and hydraulic gradient of groundwater in the basin are also controlled by the new tectonic movement. For the groundwater buried depth, generally the uplift area is larger than the fault area. The eastern tableland (10–40 m) and alluvial fan (7–15 m) have larger buried depths, while other areas are generally 1–5 m and the deeper depth can reach 5–10 m. The hydraulic gradient decreases from the uplift zone to the fault zone, the tableland ($1.111 \times 10^{-3}$) and the alluvial fan ($0.667 \times 10^{-3}$) are larger with great runoff conditions, and the low plain runoff is very sluggish ($0.109 \times 10^{-3}$). Atmospheric precipitation and leakage of the Taoer river are the main sources of recharge in the study area, and the discharge is mainly lateral runoff in the uplift area, while evaporation occurs in the fault area.

The neotectonic movement in the study area not only controls the formation of the landscape but also affects the dynamic environment of the groundwater, which changes the structure, lithology, burial, and circulation conditions of the aquifer system regularly from the edge of the basin to the center, and is a basis for regional groundwater hydrogeochemical evolution and differentiation.

### 4.1.2. Hydrogeochemical Processes

Under certain climatic and neotectonic conditions, the hydrogeochemical evolution of the groundwater system in a semi-arid basin presents typical zonal characteristics [1,5]. As the flow of water is the force of element migration, dispersion, and enrichment, starting from the groundwater dynamic environment, the recharge area (flow fast), runoff area (flow medium), and the discharge area (flow slow) are partitioned on the basis of the hydrochemistry to explain the process of hydrogeochemical evolution.

### Recharge Area

The recharge area is located in the alluvial fan and eastern tableland. The lithology is mainly gravel or sand gravel, which makes the recharge area permeability coefficient 50–200 m/day in alluvial fan and 10–30 m/day in tableland, which is high. Due to the strong terrain cutting, the tableland is relatively steep, and its hydraulic gradient (1.111‰) is larger than that in alluvial fan (0.667‰), indicating that the hydrodynamic environment and runoff conditions are better in the recharge area, and the flow is dominated by horizontal movement.

The water chemistry in this environment is dominated by leaching [23]. The results of the hydrogeochemical simulation show that the water–rock reaction is mainly the dissolution of carbonate minerals. The good hydrodynamic environment and open system are conducive to the continuous entry of $CO_2$, which strengthens its solvency. Long-term lixiviation causes the soluble minerals (chlorides/sulfates) in the aquifer to migrate into the water and be depleted, which makes the aquifer dominated by insoluble carbonates or silicate minerals [22,24]. Strong water alteration causes the chemical elements in the groundwater to remain in a migrated state, and the buried depth (7–40 m) is generally greater than the ultimate evaporation depth, causing the evaporation intensity of the phreatic water to be extremely weak (even if the pan evaporation is large) [25,26] and elements are difficult to accumulate. Therefore, the groundwater anion is dominated by $HCO_3^-$, and the cation is dominated by $Ca^{2+}$, forming $HCO_3$-Ca type water with TDS less than 0.6 g/L, and the characteristic element F < 0.6 mg/L (Figure 4).

Runoff Area

As the water enters the fault basin and flows to the front edge of the alluvial fan and tableland, the terrain becomes slower and the hydraulic gradient becomes smaller (0.318‰ and 0.624‰ in the west and east, respectively). At the same time, the sediment changes from the alluvial phase to lacustrine phase, and the lithology of the aquifer becomes sandy loam, loess loam, or fine sand, with the permeability coefficient 3–5 m/day, which hinders the horizontal movement of groundwater, reduces the buried depth of groundwater (4–15 m), enhances evaporation and concentration, and accumulates chemical elements in the groundwater to some extent.

The results of simulation show that calcite and dolomite are saturated and precipitated successively in the runoff area, accompanied with the escape of $CO_2$, but gypsum, halite, and fluorite are still in the dissolved state. The content of easily transportable elemental Na increased rapidly, while the increase in $Cl^-$ content was slow (Figure 4), indicating that the increase in $Na^+$ was caused not only by the dissolution of halite and evaporation [11,20]. As groundwater flows through the thick clay layer of the middle Pleistocene, it is prone to cation exchange, that is, $Ca^{2+}$ in water is easily adsorbed by clay particles, and $Na^+$ is desorbed into groundwater (Equation (3)). The simulation results show that the exchange amount is up to 1.82 mmol/L (calculated as $Na^+$), while the eastern runoff area is smaller due to the thinner clay layer with a cation exchange amount of 0.28 mmol/L (calculated as $Na^+$). Therefore, the water chemistry in the runoff area is relatively complicated, and there are some differences between east and west. Evaporation concentration and cation exchange cause a significant increase in readily soluble $Na^+$, while the hydrochemical type changes from $HCO_3$-Ca to $HCO_3$-Ca·Na(Na·Ca) and $HCO_3$-Na, with TDS reaching 0.3–1 g/L and $F^-$ increasing to 0.5–2.5 mg/L.

Discharge Area

In the discharge area of the low plain, the terrain is low, and the groundwater runoff is sluggish, with a permeability coefficient of 0.2–0.3 m/day and a hydraulic gradient of 0.109‰. This causes the groundwater alternate speed to be very slow, and the movement is dominated by vertical movement while horizontal movement is extremely weak. Precipitation has become the main and the only source of recharge, while the amount of evaporation is 4.33 times that of precipitation (Figure 2), and the groundwater buried depth is generally shallow (usually less than 5 m), which makes evaporation and concentration the main discharge method and hydrochemical process.

As the concentrations of groundwater, calcite, and dolomite continue to precipitate, halite, gypsum, and fluorite are generally dissolved, and fluorites are over-saturated in five samples. The readily soluble elements, Na and Cl, peak in this zone. With the increase in $Na^+$, reverse cation exchange occurs, that is, $Na^+$ in water replaces $Ca^{2+}$ adsorbed by rock (Equation (4)). The simulation results show that the exchange amount is 2.44 mmol/L and 0.86 mmol/L (calculated by $Na^+$) in the west and east, respectively. Due to the uneven surface of the low plain, the aeolian landforms such as sandy dunes, sandy ridges, and sandy hills developed well, and they are distributed with the fulje. As shown in Figure 5, slow runoff and strong evaporation cause the chemical composition and salt transport of groundwater to be significantly affected by micro-geomorphology, and the changes in water chemistry are more complicated. Groundwater was buried deeper in the sandy hill with relatively high topography, and the effect of evaporation is relatively weak; the hydrochemical type is mostly $HCO_3$-Na·Mg(Na·Ca) or $HCO_3$-Na type with TDS of 0.8–1.5 g/L and $F^-$ of 1–2.5 mg/L. Groundwater in the fulje is shallow and accompanied with strong evaporation, so the chemical type becomes $HCO_3$·Cl-Na or $HCO_3$·Cl-Na·Ca, with TDS of 1.7–4.3 g/L and $F^-$ of 1.5–4 mg/L (the highest is 7.99 mg/L).

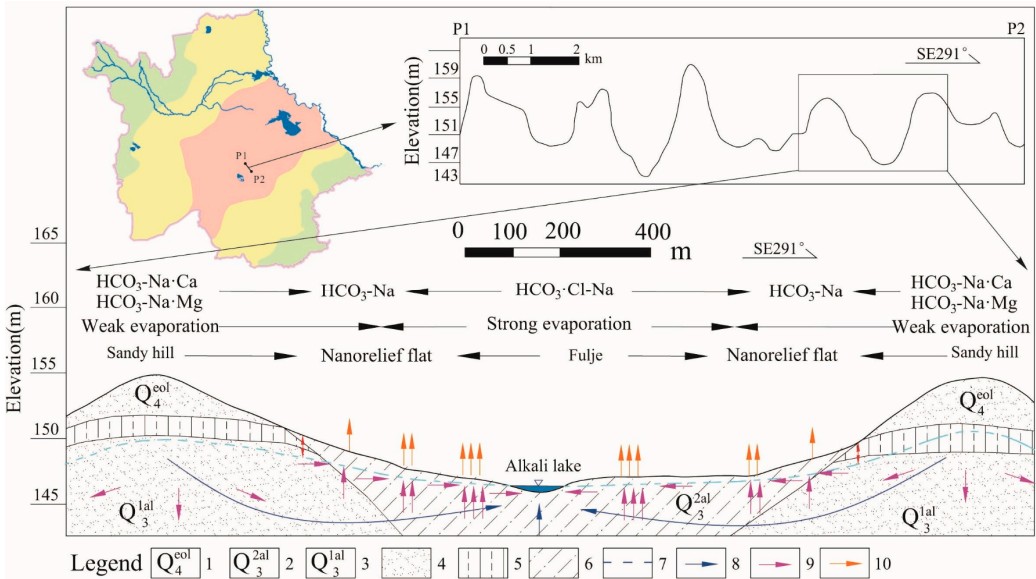

**Figure 5.** Hydrogeochemical evolution controlled by micro-topography. (1) aeolian deposit of Holocene, (2) upper part of Upper Pleistocene (lacustrine sediment), (3) lower part of Upper Pleistocene (lacustrine sediment), (4) fine sand, (5) loess loam, (6) sandy loam, (7) unconfined water level, (8) groundwater flow direction, (9) salinity transport direction, and (10) evaporation.

### 4.1.3. Characteristics of Groundwater Chemical Components

The neotectonic movement determines the spatial distribution of modern geomorphological landscapes of the basin [10]. The groundwater chemistry in different geomorphological units showing regular changes due to external forces (flowing water, sandstorm, climate). The dissolution or precipitation of the mineral and the accumulation or migration of the element caused spatial zoning of groundwater chemical components, which has the same evolutionary rules as many semi-arid basins [27].

The hydrochemistry type has typical horizontal zonation as the recharge area at the edge of the basin is mainly $HCO_3$-Ca type, occasionally $HCO_3$-Ca·Mg type, and changed from $HCO_3$-Na·Ca(Ca·Na) to $HCO_3$-Na through the runoff area. Because of the effects of evaporation, cation exchange, and micro-geomorphology, the hydrochemistry type is relatively complex, such that $HCO_3$-Na·Mg(Na·Ca) and $HCO_3$·Cl-Na (Na·Ca) type appear locally.

TDS also showed regular changes, as TDS in the recharge area was lower than 0.6 g/L, while in the discharge area it was generally higher than 0.8 g/L, and a few were higher than 1.5 g/L in the low-lying fulje under the micro-geomorphic effect. As Table 4 shows, 15.38%, 20.00%, and 39.53% of sample TDS in the recharge, runoff, and discharge areas, respectively, exceeded the standard of drinking water (the limit of TDS is 1 g/L in Sanitary Standard for Drinking Water of China).

Trace element F, which is closely related to human health, also exhibits change on a regular basis. In the recharge area, where the groundwater alteration is strong, F is easy to migrate and accumulate slowly, with a content less than 0.6 mg/L. When the groundwater reached the runoff zone, F begins to accumulate, as the content is greater than 0.5 mg/L. In the central discharge area of the basin, the strong evaporation and stagnation flow cause the elements to accumulate rapidly. The alkaline water environment (average of pH is 7.69) and the soda type of water chemistry are also conducive to the enrichment of F [28,29], finally forming high-fluorine water, which is not suitable for drinking. As Table 4 shows, 20.51%, 56.67%, 67.44% of sample F in the recharge, runoff, and discharge area, respectively, exceeded the standard of drinking water (the limit of F is 1 g/L in Sanitary Standard for Drinking Water of China).

**Table 4.** The over-standard of TDS and F in each zone.

| | Items | Recharge Area (n = 39) | Runoff Area (n = 60) | Discharge Area (n = 43) | Total (n = 142) |
|---|---|---|---|---|---|
| | Average of pH | 7.59 | 7.67 | 7.69 | 7.65 |
| TDS | Number of over standard | 6 | 12 | 17 | 35 |
| | Percentage | 15.38% | 20.00% | 39.53% | 26.65% |
| F | Number of over standard | 8 | 34 | 29 | 71 |
| | Percentage | 20.51% | 56.67% | 67.44% | 40.80% |

### 4.2. Quantitative Analysis of Using Different Indicators

The differentiation of groundwater chemical characteristics in different zones can be explained and characterized by different indicators.

#### 4.2.1. Hydrodynamic Indicator

As the carrier of dissolved minerals, water flow is the basic force for the migration, dispersion, and accumulation of elements in the groundwater. The change in the hydrodynamic conditions is caused by the friction of the surface of the particles when the water flows in the porous medium [1,30]. Therefore, the hydrodynamic index can reflect the time or extent of the interaction between the groundwater and porous medium.

The hydraulic gradient (0.667–1.111‰) and permeability coefficient (15–200 m/day) in the recharge area are much larger than that in the discharge area (hydraulic gradient: 0.109‰, permeability coefficient: 0.2–0.3 m/day) (Table 3), reflecting a relatively good hydrodynamic environment and strong water alteration. In addition, $\gamma Cl^-/\gamma Ca^{2+}$ is one of the parameters that characterizes the hydrodynamics, as $Cl^-$ is usually enriched in the stagnant region of groundwater and the cations in the rapid water flow are dominated by $Ca^{2+}$. Therefore, high $\gamma Cl^-/\gamma Ca^{2+}$ represents slow hydrodynamics condition. The increase in $\gamma Cl^-/\gamma Ca^{2+}$ in the study area from the recharge zone to the runoff zone was slow, while it increased rapidly from the runoff zone to the discharge zone (Figure 6), reflecting the change in the hydrodynamic conditions from strong to weak, and the discharge zone was much weaker than that in the recharge and runoff area. Changes in regional hydrodynamic indicators and water chemistry indicate that the two are not independent but interrelated [1,31], especially in the semi-arid water storage basin, where a good hydrodynamic environment is characterized by elemental migration and low TDS water, while a poor hydrodynamic environment is characterized by elemental enrichment and higher TDS water.

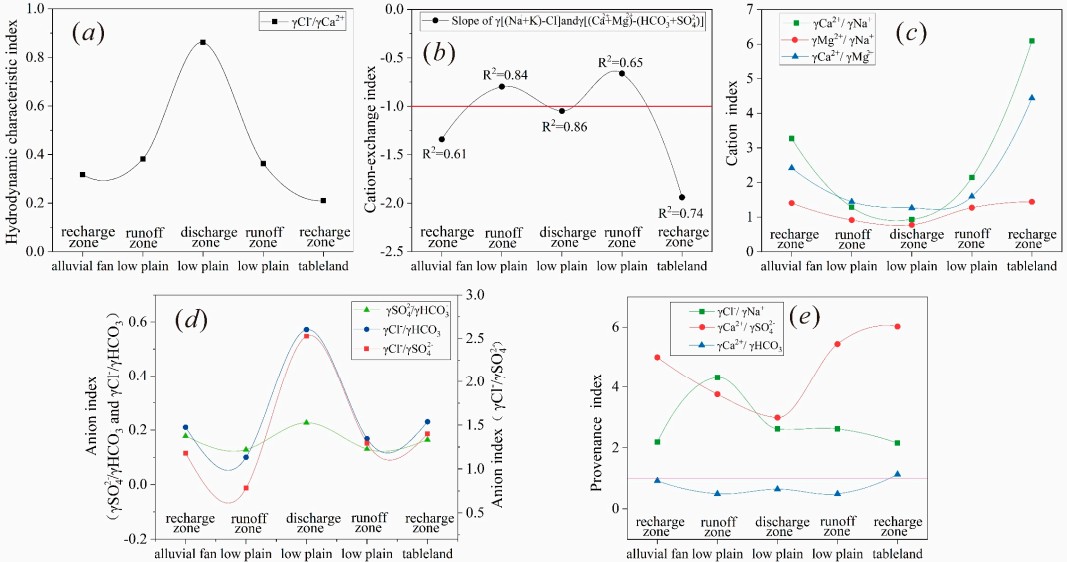

**Figure 6.** Quantitative index in different partitions (**a**) $\gamma Cl^-/\gamma Ca^{2+}$, (**b**) slope of $\gamma(Na^+ + K^+ - Cl^-)$ and $\gamma(Ca^{2+} + Mg^{2+}) - (SO_4^{2-} + HCO_3)$, (**c**) cation ratio coefficient, (**d**) anion ratio coefficient; (**e**) provenance index

### 4.2.2. Cation Exchange Index

Cation exchange is actually an adsorption–desorption process, and the occurrence possibility can be indicated by the relationship between $\gamma(Na^+ + K^+ - Cl^-)$ and $\gamma(Ca^{2+} + Mg^{2+}) - (SO_4^{2-} + HCO_3)$ [20,32]. Except for the dissolution of halite, $\gamma(Na^+ + K^+ - Cl^-)$ indicates the increase or decrease of $Na^+$, while $\gamma(Ca^{2+} + Mg^{2+}) - (SO_4^{2-} + HCO_3)$ indicates the increase or decrease of $Ca^{2+}$ and $Mg^{2+}$, except for the dissolution of calcite, dolomite, and gypsum. When the cation exchange is significant, the relationship between the two is linear and the slope is close to $-1$. However, this parameter can only reflect the possibility of cation exchange but not the direction of exchange. As shown in Figure 6b, the western runoff zone (k = $-0.8$, $R^2$ = 0.84) and the discharge zone (k = $-1.05$, $R^2$ = 0.86) have a good linear relationship, indicating significant cation exchange, while the recharge zone and the eastern runoff zone do not.

According to [33], the clay minerals in the study area are mainly montmorillonite and illite, and the content of kaolinite is relatively small. As the montmorillonite (80–150 meq/100 g) and illite (10–40 meq/100 g) have the bigger CEC (Cation Exchange Capacity) and the CEC of kaolinite is only 3–15 meq/100 g [34], the clay minerals in the basin have strong cation exchange capacity. The alkaline pH (Table 1) in the basin can increase the variable negative charge in the clay and increase its CEC value [9,35]. Therefore, the strength of the cation exchange reaction in the basin is controlled by the distribution in the clay layer (Figure 4), which is more significant in the western runoff area and the middle discharge area with a thicker clay layer and consistent with the results of the cation exchange index (Figure 6b) and reverse simulation (Table 2).

### 4.2.3. Ion Ratio Coefficient

The ion ratio coefficient can be used to determine the origin of groundwater or the source or formation process of the groundwater chemical component [20,36]. The cation ratio $\gamma Ca^{2+}/\gamma Na^+$, $\gamma Mg^{2+}/\gamma Na^+$, and $\gamma Ca^{2+}/\gamma Mg^{2+}$ showed a regular downward trend from the recharge to the discharge zone, which indicated the gradual enrichment of $Na^+$ and $Mg^{2+}$ in groundwater. The migration order of the major cation is $Na^+ \rightarrow Mg^{2+} \rightarrow Ca^{2+}$, instead of the generally considered $Na^+ \rightarrow Mg^{2+} \rightarrow Ca^{2+}$ [34], which may be related to significant cation exchange. The authors of [37] also reached the parallel conclusion in the study of groundwater chemistry in the arid region.

Different from cations, anions $\gamma Cl^-/\gamma HCO_3^-$ and $\gamma SO_4^{2-}/\gamma HCO_3^-$ are all less than 1, and the trend from the recharge to the runoff zone shows a decreasing trend, indicating that the leaching effect is still significant, causing the increase in $HCO_3^-$ to be greater than that in $Cl^-$ or $SO_4^{2-}$. When the groundwater enters the discharge area, the content of $HCO_3^-$ reaches the peak without continuous increase (Figure 4), which is attributed to the saturation state of calcite and dolomite, while $Cl^-$ and $SO_4^{2-}$ can continue to accumulate because of the unsaturated state of halite and gypsum, so that $\gamma Cl^-/\gamma HCO_3^-$ and $\gamma SO_4^{2-}/\gamma HCO_3^-$ are increased. The change in $\gamma Cl^-/\gamma SO_4^{2-}$ indicates that the difference between the content of $Cl^-$ and $SO_4^{2-}$ in the recharge and runoff regions is small, and the cumulative velocity of $Cl^-$ is faster than that of $SO_4^{2-}$ in the discharge zone, but relatively speaking, $HCO_3^-$ is still the dominant anion.

The value of $\gamma Na^+/\gamma Cl^-$ in the recharge zone is greater than 1, that is, $\gamma Na^+$ is generally higher than $\gamma Cl^-$, which indicates that the dissolution of halite is not the only source of $Na^+$, and the remaining $Na^+$ may be derived from the weathering and dissolution of aluminosilicate, such as albite (Equation (5)) and sodium montmorillonite (Equation (6)). The abnormal increase in $\gamma Na^+/\gamma Cl^-$ in the western runoff region may be the result of cation exchange (Equation (1)), while the decrease in the discharge zone may be affected by reverse cation exchange (Equation (4)).

$$4NaAlSi_3O_8 + 4CO_2 + 22H_2O \rightarrow Al_4(Si_4O_{10})(OH)_8 + 8H_4SiO_4 + 4Na^+ + 4HCO_3^- \tag{5}$$

$$3Na_{1/3}Al_{7/3}Si_{11/3}O_{10} + 30H_2O + 6OH^- \rightarrow Na^+ + 7Al(OH)_4^- + 11H_4SiO_4 \tag{6}$$

The ratio $\gamma Ca^{2+}/\gamma HCO_3^-$ is less than 1, indicating that the dissolution of dolomite is also an important source of $Ca^{2+}$. The trend of decrease first then increase confirms the existence of

the positive ion exchange in the runoff zone and the reverse cation exchange in the discharge zone. The value of $\gamma Ca^{2+}/\gamma SO_4^{2-}$ decreased significantly from the recharge zone to the discharge zone, indicating that the enrichment of $SO_4^{2-}$ is greater than that of $Ca^{2+}$. As Figure 7 shows, gypsum is always in dissolved mode while calcite and dolomite undergo precipitation, indicating that the enrichment of $Ca^{2+}$ may be limited by the cation exchange (Equation (3)) and the precipitation of calcite and dolomite (Equations (1) and (2)).

### 4.2.4. Thermodynamic Index

Weathering is the basis of groundwater chemical formation, which is reflected in the dissolution and precipitation of minerals in the water–rock interaction. The thermodynamic index of the ability of groundwater to dissolve minerals is usually expressed by the saturation index of minerals.

Figure 7 shows the tendency of the calcite and dolomite in the groundwater to gradually evolve from unsaturated to saturated from the recharge zone to the discharge zone. A considerable number of water samples have been found to be over-saturated in the recharge zone, indicating that the transition from unsaturated to saturated occurs in the recharge zone. In addition, there are still water samples in the runoff zone and the discharge zone that are unsaturated, which may be related to cation exchange and the microtopography effect.

For halite and gypsum, all water samples are in an unsaturated state, and the saturation index is between −3.24 and −0.18 and between −9.21 and −4.95, as it tends to increase from the recharge zone to the discharge zone, indicating that halite and gypsum are continuously dissolved. The unsaturation state of halite depends on its large solubility of chlorides, while the unsaturation state of gypsum is related to the alluvial and lacustrine aquifers, as there are relatively few sulfate minerals in the sedimentary facies [22,27].

For fluorite, only five water samples are saturated, and all of them are located at the discharge zone, where fluorite precipitation occurs. The elevation of these five sampling points is 125–136 m, which is significantly lower than the elevation of other sampling points (138–157 m) in the discharge area, that is, the sampling point is located at the low-lying area. Therefore, the dissolution and precipitation of fluorite, and the enrichment of F in the discharge area are mainly controlled by the microgeomorphology effect.

The calculation results of the saturation index of the study area are consistent with the results of the hydrogeochemical simulation. The state of minerals in the groundwater is also consistent with other semi-arid water storage basins, such as the Yinchuan Plain of China [4] and Weining Plain of China [37], Central Tunisia [6], and Hamersley Basin of northwest Australia [38]. Calcite and dolomite from the recharge zone to the discharge zone change from unsaturated to saturated, gypsum and rock salt are generally unsaturated, and fluorite is saturated only in the discharge area.

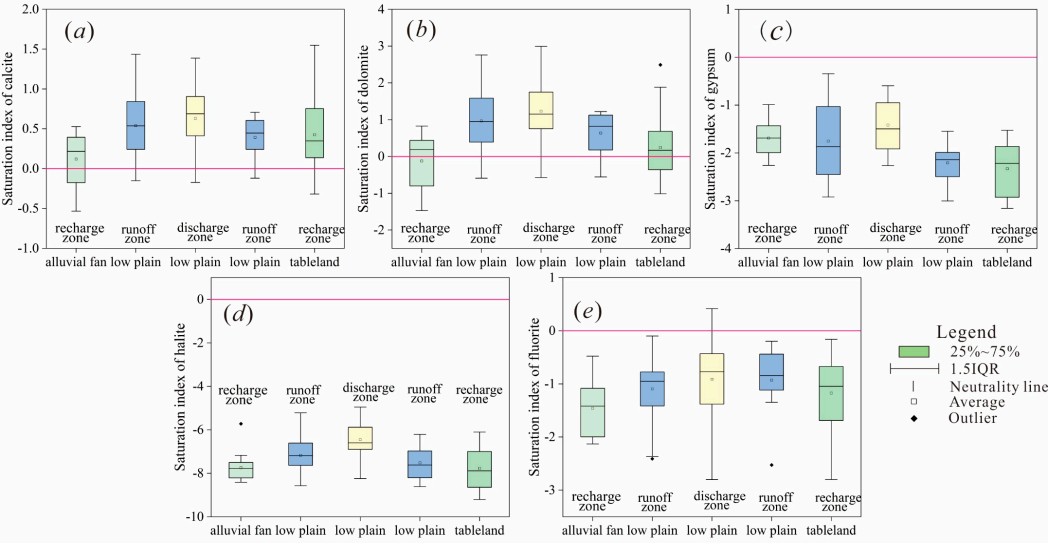

**Figure 7.** Saturation index in different partitions: (**a**) Calcite (**b**) Dolomite (**c**) Gypsum (**d**) Halite (**e**) Fluorite.

## 5. Conclusions

(1)  Taking the western part of Jilin Province as an example, a hydrogeochemical evolutionary zoning model of a typical semi-arid water storage basin was established. The model is divided into three layers from bottom to top. The first layer represents the geological and hydrogeological conditions, including the topography, lithology, geological time, and hydrodynamic characteristics. The second layer reveals the hydrogeochemical processes, divided into the recharge zone, runoff zone, and discharge zone in the horizontal direction. The third layer is the characteristics of the groundwater chemical components, including chemistry type, TDS, main anion and cation, and characteristic element F. This zoning model shows that the hydrogeochemical action gradually changes from lixiviation to cation exchange, evaporation controls the hydrogeochemical evolution, and landform plays a key role in hydrochemistry formation in the discharge area.

(2)  The quantitative analysis was carried out by hydrogeochemical reverse simulation. The results showed that from the recharge to the discharge zone, calcite and dolomite change from unsaturated to saturated, synchronized with the entry and escape of $CO_2$ from the groundwater. Fluorite, gypsum, and halite are always in a dissolved state. The evaporation is gradually enhanced; the cation exchange changed from $Na^+$ released into the water to $Ca^{2+}$ released into the water. The differences in the lithology of the formation make the exchange amount quite different in the runoff area of the east and west.

(3)  The hydraulic gradient, permeability coefficient, and $\gamma Cl^-/\gamma Ca^{2+}$ indicate that the change in the hydrodynamic environment worsens from the recharge zone to the discharge zone. The cation exchange index proves that the cation exchange is more significant in the western runoff zone and the discharge zone, while $\gamma Ca^{2+}/\gamma Na^+$, $\gamma Mg^{2+}/\gamma Na^+$, and $\gamma Ca^{2+}/\gamma Mg^{2+}$ show that the migration order of cations from the recharge zone to discharge zone is $Na^+ \rightarrow Mg^{2+} \rightarrow Ca^{2+}$. The $\gamma HCO_3^-/\gamma Cl^-$. $\gamma SO_4^{2-}/\gamma Cl^-$, $\gamma SO_4^{2-}/\gamma HCO_3^-$ results show that from the recharge to the runoff zone, the content of $HCO_3^-$ shows a decreasing trend while reaching the peak without continuous increase in the discharge area, which is attributed to the saturation state of calcite and dolomite. Meanwhile, $HCO_3^-$ is still the dominant anion in groundwater, and $\gamma Na^+/\gamma Cl^-$, $\gamma Ca^{2+}/\gamma HCO_3^-$, and $\gamma Ca^{2+}/\gamma SO_4^{2-}$ indicate the source of $Na^+$ and $Ca^{2+}$, demonstrating the positive cation exchange in the western runoff zone and the reverse cation exchange in the discharge zone. The saturation indexes indicate that the calcite and dolomite from the recharge zone to the discharge zone change from unsaturated to saturated, gypsum and rock salt are generally unsaturated, and fluorite is saturated only in the discharge area, which is consistent with the results of the reverse simulation and consistent with the other semi-arid reservoirs, as well as with the results of hydrogeochemical simulation and other semi-arid water storage basins.

**Author Contributions:** M.L., S.H., and Y.C. processed the data and analyzed the results; M.L. wrote the manuscript; Y.C, X.L, and C.X. reviewed the manuscript and made helpful suggestions; M.L. revised the manuscript.

**Funding:** The study was financially supported by Natural Science Foundation of China (No. 41572216), the China Geological Survey Shenyang Geological Survey Center "Changji Economic Circle Geological Environment Survey" project (1212010070000150012), the Provincial School Co-construction Project Special-Leading Technology Guide (SXGJQY2017-6), and the Jilin Province Key Geological Foundation Project (2014–13).

**Acknowledgments:** We would like to thank the anonymous reviewers and the editor.

**Conflicts of Interest:** The authors declare no conflict of interest.

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
