# Peer review of "Hydrochemical Evolution of Groundwater in a Typical Semi-Arid Groundwater Storage Basin Using a Zoning Model"

_water, doi:10.3390/w11071334_

Round 1
Reviewer 1 Report
The manuscript title " Hydrochemical evolution of groundwater in a typical semi-arid groundwater storage basin using a zoning" well-written manuscript, except a few typing error and grammar mistake. I check the Plagriosm and it is not at all an issue, Plagiarism report is attached as an attachment here. Manuscript fulfils the journal standards and recommends to accept the manuscript.
Line No 73 it should be “the environment” instead environment
Line No 79 it should be “was” instead were
Line No 148 it should be “was” instead were
Line No 158 it should be “is” instead are
Line No 228 it should be “is” instead are
Line No 254 it should be “in a different” instead in different
Line No 335 it should be “cause a significant” cause significant
Line No 389 it should be “of sample” instead of the sample
Author Response
Response to Reviewer 1 Comments
Point 1: Line No 73 it should be “the environment” instead environment
Line No 79 it should be “was” instead were
Line No 148 it should be “was” instead were
Line No 158 it should be “is” instead are
Line No 228 it should be “is” instead are
Line No 254 it should be “in a different” instead in different
Line No 335 it should be “cause a significant” cause significant
Line No 389 it should be “of sample” instead of the sample
Response 1: We quite appreciate your consideration and the comments concerning our manuscript entitled “Hydrochemical evolution of groundwater in a typical semi-arid groundwater storage basin using a zoning model” (ID: water-521969). Now, all the typing error and grammar mistake were revised, and the "Track Changes" function in Microsoft Word was used, so that reversions are easily visible to you. We hope this revision can make the paper more acceptable.

Reviewer 2 Report
This manuscript is very well written. An appropriate number of samples was collected and many chemical analyses were preformed on these 142 samples. I didn't see any altitudes of the study area given, but general water flow was pointed out as "controlled by topography". Except for the use of FHREEQC program, I assume all other programs were author-created or in the referenced articles. It was not clear to me.
1. The authors have an amazing amount of results (data) from their many analyte measurements on 142 samples of water. There are no tables of these experimental data: concentrations of anions, cations, pH, temperature, etc. In any experimental investigation the data is the most important information. Other scientists can use the data and make other comparisons.
2. With the tables of data, not presented, the authors should try Principle Component Analysis (Cluster Analysis). PCA(Principal Component Analysis) often produces very unexpected and important results.
Author Response
Response to Reviewer 2 Comments
Point 1: I didn't see any altitudes of the study area given, but general water flow was pointed out as "controlled by topography".
Response 1: The topographic map of the study area can be seen in Figure 1(b). Similarly, the elevation data is also displayed in the zoning model of Figure 4 (at the y-axis of the lower left corner). The elevation column in Table 2 also descripts of the elevations in different zones.
Point 2: The authors have an amazing amount of results (data) from their many analyte measurements on 142 samples of water. There are no tables of these experimental data: concentrations of anions, cations, pH, temperature, etc. In any experimental investigation the data is the most important information. Other scientists can use the data and make other comparisons.
Response 2: We added the statistics of groundwater chemical data in each hydrogeochemical zone as you suggested (Table 1), at the same time, the serial number of the tables has been adjusted accordingly.
Point 3: With the tables of data, not presented, the authors should try Principle Component Analysis (Cluster Analysis). PCA (Principal Component Analysis) often produces very unexpected and important results.
Response 3:
Multivariate statistical techniques such as hierarchical cluster analysis (HCA) and principal components analysis (PCA) of the groundwater chemistry data can provide a new view of hydrochemical evaluation as the reviewer suggested. The results of HCA help in interpreting the data and indicate patterns and PCA has been applied successfully to extract variables and infer the underlying natural and/or anthropogenic processes that control the chemistry of groundwater. However, our another manuscript entitled“Impact of anthropogenic and natural processes on the evolution of groundwater chemistry in a typical semi-arid groundwater storage basin” (ready to submit) is using the method you mentioned (PCA and HCA). In order to avoid duplication of the content of these two articles and cause unnecessary misunderstanding, we decided not to use these two methods in this article.
Special thanks to you for your good comments.
Reviewer 3 Report
It is an excellent paper that describes a hydrogeochemical evolutionary zoning model of groundwater basin of the semi-arid area in China. The concept of the model can be applied worldwide for the same type of groundwater basins and climate conditions. The concept of the paper is clear and authors applied innovative and state of the art methods. Application of hydrogeochemical simulation model, using PHREEQC, is very well documented and properly described. The paper does not need any changes before publication. I suggest to publish it in the present form.
Author Response
Thank you for your appraisals concerning our manuscript entitled “Hydrochemical evolution of groundwater in a typical semi-arid groundwater storage basin using a zoning model” ,we really appreciate.